# Glaucoma Treatment and Hydrogel: Current Insights and State of the Art

**DOI:** 10.3390/gels8080510

**Published:** 2022-08-17

**Authors:** Antonio Maria Fea, Cristina Novarese, Paolo Caselgrandi, Giacomo Boscia

**Affiliations:** Department of Ophthalmology, University of Turin, Via Cherasco 23, 10126 Turin, Italy

**Keywords:** hydrogel, drug delivery, glaucoma, nanoparticles, ophthalmology, hydrogel in ophthalmology, glaucoma therapy, hydrogel in glaucoma, glaucoma treatment

## Abstract

Aqueous gels formulated using hydrophilic polymers (hydrogels) and those based on stimuli-responsive polymers (in situ gelling or gel-forming systems) attract increasing interest in the treatment of several eye diseases. Their chemical structure enables them to incorporate various ophthalmic medications, achieving their optimal therapeutic doses and providing more clinically relevant time courses (weeks or months as opposed to hours and days), which will inevitably reduce dose frequency, thereby improving patient compliance and clinical outcomes. Due to its chronic course, the treatment of glaucoma may benefit from applying gel technologies as drug-delivering systems and as antifibrotic treatment during and after surgery. Therefore, our purpose is to review current applications of ophthalmic gelling systems with particular emphasis on glaucoma.

## 1. Introduction

A progressive loss of the visual field and eventually blindness characterize glaucoma, an optic neuropathy caused by the destruction of ganglion cells [1]. Glaucoma is the second most common cause of blindness worldwide, with a growing number of people affected [2,3]. Glaucoma can dramatically compromise individuals’ habits and quality of life (QoL), determining a progressive and irreversible visual impairment and, potentially, blindness [4].

Intraocular pressure (IOP) reduction is currently the mainstay of glaucoma therapy, and it may be obtained with drugs, laser, or surgical therapy [5,6,7]. The first-line treatment is presently based on topical drug treatment.

Multiple observational studies and randomized clinical trials have confirmed that antiglaucoma medications effectively lower IOP at various disease stages [7,8,9,10,11]. Beta-blockers, topical carbonic anhydrase inhibitors, cholinergics, prostaglandins/prostamides, alpha2-agonists, and Rho-kinase inhibitors are only a few medication families used alone or in combination to treat glaucoma. Though medical therapy is a crucial asset for patients and ophthalmologists, it can have significant drawbacks. For example, adverse effects (ocular inflammation, itching, and visual disturbances) reduce adherence, thereby diminishing the therapeutic efficacy. In addition, it is challenging for patients and caregivers to administer eye drops; administration is frequently suboptimal. For these reasons, the adherence to glaucoma medical therapy is often poor, with an early interruption of the treatment [12,13]. Moreover, topical treatments may harm conjunctival health, reducing the surgery’s effectiveness [12,13].

Furthermore, the eyes have several metabolic, static, and dynamic barriers that increase the difficulty of topical drug administration. The conjunctiva, sclera, and cornea are the static obstructions that medications must get through to work pharmacologically on the eye. These barriers are challenging because of their various characteristics (polarity, hydrophilicity, surface charge, etc.).

On the contrary, dynamic barriers are the rapid clearance of the drops from the eye through nasolacrimal drainage and conjunctival vessels. Moreover, metabolic enzymes of the ocular surface can deactivate drugs [14]. The sum of these mechanisms drastically reduces the pharmaceuticals’ absorption due to the lower residence time on the ocular surface. Reduced effectiveness of the treatment leads to frequent changes in the therapy and, thus, to poorer patient compliance.

Conventional ophthalmic dosages, formulated as eye drops (solutions or suspensions), or ointments, are preferable to distribute medications to the ocular surface and anterior eye segment. Their comparatively low production costs, non-invasiveness, and ease of use offer clear benefits. [15,16].

Their side effects and low bioavailability may reduce optimal drug concentration at the target site (Figure 1). To avoid the failure of the medical treatment, several researchers have developed various drug delivery systems to enhance the residence time on the ocular surface [17,18,19]. One of the most frequently adopted strategies is medical gels [20].

Another exciting field of gel application is to prevent scarring after filtering glaucoma surgery.

This review will focus on drug delivery and adjunctives in glaucoma surgery.

### Polymers and Hydrogel in Ocular Drugs Delivery

Hydrogels are a network of biologic and synthetic hydrophilic polymers able to absorb aqueous fluids. The extensive array of customizability and the high hydrophilicity render hydrogel the ideal compound for ophthalmic applications, ranging from vitreous substitutes to drug delivery [21]. Several authors proposed hydrogel as a drug carrier, incorporated with other polymers, to treat glaucoma. Among these molecules, synthetic polymeric biomaterials are a group of compounds based on chemically-derived monomers and represent one of the most reported options for delivering ocular pharmaceuticals [21]. To this class belong polyethylene oxides, polyvinyl alcohol (PVA), polyesters, polymethacrylates, polyolefins, and dendrimers [22].

On the contrary, biopolymers are based on naturally derived monomers and are characterized by high biocompatibility and fast degradation [23]. The most frequently reported biological polymers used for ocular drug delivery are chitosan, cellulose, hyaluronic acid, carboxymethyl cellulose, and gelatin. Another promising field in ocular delivery systems is micro- and nanotechnology, such as formulations of hydrogel combined with micro- and nanoparticles, liposomes, and micelles [22]. This drug delivery strategy combines the hydrogel’s high water content with the small size (from 1 to 1000 nm) of these particles, offering a more targeted delivery and a sustained release of the medications [23].

Among the medical gels, the most investigated are the gel-forming systems. Their properties make them able to undergo phase transition after environmental stimulus following their application in the conjunctival cul-de-sac and thus, transform themselves into a viscoelastic gel from the original liquid dosage [24] because of pH change, temperature modulation, or ion triggers [25,26,27,28,29,30]. Because they do not need organic solvents or copolymerization catalysts to cause gel formation, in situ polymeric gelling systems have drawn significant attention.

## 2. Drug Delivery

The use of topical medications represents the primary approach to treating glaucoma. Several therapeutic classes, α-2 agonists, β-blockers, carbonic anhydrase inhibitors, prostaglandin analogous, and cholinergic agents, were developed in the form of eye drops for IOP regulation [31].

A correct administration of eye drops and proper therapy can prevent an IOP elevation and, thus, the progression of the glaucoma-induced damages.

Despite that, as stated above, self-administration of topical therapy can be challenging for an elderly population. Moreover, ophthalmic solutions are often related to local side effects such as ocular burning and stinging [32,33,34,35]. These conditions may lead to poorer therapy compliance and, thus, can cause a worse treatment outcome [36,37]. Therefore, improving patient adherence and compliance while facilitating the use of glaucoma drugs is a top public health priority [38,39].

Furthermore, drugs’ efficacy may be decreased due to their rapid clearance (low residence time), which results in a lower bioavailability [17,40]. For these reasons, a proper delivery system may help improve medications’ effectiveness, safety, and duration.

Then, several electronic devices, transdermal and ocular inserts, and mechanical drug delivery systems were developed [41].

### 2.1. Contact Lens

Due to their location near the cornea, contact lenses (CL) have some unique advantages for delivering drugs [42]. First of all, the limitation opposed by the CL to the tear film action results in a drug residence time of more than 30 min [43,44] compared to 5 min only for eye drops alone [45]. The enhanced residence time leads to significant increases in bioavailability, possibly as large as 50% [46].

Notably, hydrogels are the main constituents of CL; thus, their high-water content and favorable properties render them highly compatible with human tissues [47,48,49]. Indeed, even if the water content of the lens reaches 99%, the oxygen permeability will not exceed the theoretical value of about 40 DK/t, which affects the user’s comfort during a prolonged period [50].

For these reasons, CL can be used to treat ocular diseases such as glaucoma [51] due to their ability to extend the release of a preloaded drug to several days or even months [52].

These drugs can be loaded into lenses by soaking [53,54], molecular imprinting [55], microemulsion, or through nanoparticles [56,57].

Current methods can produce lenses of suitable thickness, water content, and optical properties [58,59]. Moreover, it was shown that the application time did not affect ocular tissues, and no ocular adverse effects were observed in any case [60].

Furthermore, therapeutic CL can reduce drug doses and thus side effects [61] and can be particularly beneficial for applications in elderly patients having trouble adhering to repeated dosage regimens [62,63]. CL can be made of hydroxyethyl methacrylate (HEMA), Methacrylic acid (MAA) [55], poly(vinyl alcohol) (PVA) [64], and N-vinyl-2-pyrrolidone (NVP), Poly(2-hydroxyethyl methacrylate) (pHEMA) [65]. Moreover, some authors, such as Gulsen et al. [66,67,68] and Kapoor et al. [69,70], have proposed CL incorporated with nanoparticles, such as liposomes, micelles, and microemulsions [71,72,73,74,75,76,77,78,79]. Jung et al. focused on soaking commercial lenses in a solution of timolol-Propoxylated Glyceryl Triacylate (PGT) nanoparticles or timolol- Ethylene Glycol Dimethacrylate (EGDMA) nanoparticles. Drug release studies in a diffusion cell showed an extended release for about 2–4 weeks [80].

Maulvi et al., in 2016, formulated a nanoparticle-loaded ring implant placed between partially polymerized hydrogel contact lenses to provide an extended release of Timolol Maleate (TM) at therapeutic rates without affecting the optical or physical properties of CL. In vivo studies showed a tear fluid release for more than 192 h [81] (Figure 2).

Subsequently, Maulvi et al., in 2021, proposed to use graphene oxide (GO) loaded into silicone hydrogel CL through a polymerization process to improve the Bimatoprost release. GO improved the swelling properties of the lenses, the transmittance, and the bioavailability [82].

Ciolino et al. created Latanoprost-eluting contact lenses by encapsulating latanoprost-PLGA (poly[lactic-co-glycolic acid]) between two layers of pHEMA hydrogel by ultraviolet light polymerization. In vitro and in vivo studies reported an extended release of up to one month [83].

Sekar et al. showed the ability of the vitamin-E-integrated polymeric hydrogel to prolong prostaglandins (Bimatoprost) release for more than ten days [84], while increasing stability and limiting diffusion during storage [85,86]. Indeed, vitamin-E incorporation into the hydrogel matrix has been reported to form nano-sized barriers shaped as ellipsoids, retarding drug diffusion through the polymer matrix [84].

Nicolson et al. found out that hydrogel lenses based on hydrophilic monomers such as 2- HEMA and NVP did not succeed in absorbing the minimum oxygen requirement for eyes under closed eyelid conditions [87].

Silicone hydrogel CL offers advantages over other types of lenses in drug delivery because of their oxygen transmissibility and their potential for hydrophobic drug delivery [88].

Nguyen-Phuong-Dung et al. showed that higher content of hydrophilic polymers increased water uptake ability and improved hydrophilicity of silicone hydrogel lenses. However, the oxygen permeability is linked with the quantity of polydimethylsiloxane (PDMS): the permeability decreases with the decrease of PDMS content. In addition, these silicone hydrogel lenses exhibited relatively good optical transparency, anti-protein deposition and appeared non-cytotoxic [89].

Chen et al. and Jones et al. explained that PDMS exhibits a lower water uptake ability, poor wettability, and high lipid adsorption despite having unique high oxygen transmissibility and superior resistance to tearing for contact lens application [87,90].

One advantage of the hydrophobic drug released from CL is that it could more readily diffuse through corneal cells’ tight lipid junctions and cell membranes due to its hydrophobic character.

Furthermore, different studies have found that drugs with hydrophobic character (ex. Latanoprost) can be loaded quickly (in 4 min) into Hydrogel contact lenses using non-aqueous solvents [91].

One of the most critical problems of drug delivery through CL remains the difficulty of incorporating more than one drug. A high degree of cross-link hydrogel is needed in this case, but this process might alter physical properties or oxygen permeability [92,93,94,95] (Table 1).

### 2.2. Hydrogel as an Ocular Drug Delivery System for Glaucoma Treatment

As stated above, the essential properties of hydrogels are their ability to absorb water due to their hydrophilicity and excellent biocompatibility, making them ideal for drug delivery applications [96]. Several authors proposed hydrogel as a drug carrier incorporated with other polymers such as chitosan, cellulose derivatives, gelatin, PVA, or poly lactic-co-glycolic acid for the treatment of glaucoma [96,97,98,99]. Other polymer-based hydrogels evaluated for ophthalmic administration are stimulus-responsive gel [100,101,102,103,104]. These delivery strategies have been investigated for several ophthalmic drugs commonly used in the treatment of glaucoma, such as TM, brimonidine tartrate (BT), pilocarpine, and latanoprost, or combinations with each other [105,106,107,108] (Table 2).

#### 2.2.1. Pilocarpine

Pilocarpine is a muscarinic acetylcholine M3 agonist, causing miosis and increasing aqueous humor outflow [109]. In 2007, Natu et al. developed, as drug carriers, hydrogels loaded with pilocarpine hydrochloride by soaking in an aqueous solution containing the drug [110]. In vitro evaluation showed a percentage of released drugs that varied between 29.2% and 99.2% in 8 h [110]. Subsequently, Chou et al. developed pilocarpine-loaded antioxidant-functionalized biodegradable thermogels for intracameral administration of antiglaucoma medications [107]. The antioxidant function is due to the chemical grafting of antioxidant gallic acid onto biodegradable gelatin [107]. In vivo study on a rabbit model showed a mean IOP reduction of 5 mmHg for 28 days [107].

Lai and co-workers proposed a new biodegradable in situ gelling delivery system for the intracameral administration of the pilocarpine [111]. The preparation of the copolymeric carriers was achieved from gelatin-g-poly(N-isopropyl-acrylamide). All prepared copolymeric carriers exhibit a relevant intraocular pressure-lowering and an excellent miotic effect [111]. Afterward, Nguyen and colleagues prepared injectable biodegradable thermogels coloaded with pilocarpine and ascorbic acid [112]. The synthesized gel obtained the double effect of lowering elevated IOP and reducing stromal collagen degradation in inflammation-induced glaucoma. In vitro results indicated a sustained release for 80 days [112]. In 2020, Luo et al. reported a new biodegradable and injectable thermogel, coloaded with pilocarpine and RGFP966 to exert antioxidant activities and sustained drug delivery for treating glaucomatous nerve damage [113]. RGFP966 is an inhibitor of histone deacetylases and, consequently, plays a critical role in regulating retinal ganglion cell atrophy and is becoming a primary target for treating neurodegeneration [113]. Their results indicate a long-active release function of both drugs [113].

#### 2.2.2. Timolol Maleate

TM is a small hydrophilic molecule (432 Da) regarded as the “gold standard” treatment for glaucoma [114,115]. Indeed, the intraocular pressure (IOP)-lowering potential of this β-receptor antagonist has been reported to be between 20 and 25% of the initial values [116]. Incorporating viscosifying agents can avoid some limitations of topical administration, such as extensive drug loss due to the turnover of lacrimal drainage. Several authors proposed hydrogel-based formulation for TM application [117]. In 2011 Zhang et al. investigated a novel TM liposomal-hydrogel to improve drug permeability and extend residence time in the precorneal region [118]. In an in vivo study on 12 rabbits, it lowered IOP for six h before rising to its initial value [118]. Later, in 2012 Holden et al. proposed a polyamidoamine dendrimer hydrogel linked with polyethylene glycol (PEG)-acrylate chains for BT and TM delivery [119]. This dendrimer hydrogel increased the solubility of brimonidine and sustained the in vitro release of both drugs over 56–72 h [119].

Yang et al. developed a hybrid dendrimer hydrogel/poly(lactic-co-glycolic acid) nanoparticle platform to efficiently deliver BT and TM to the eye and gradually release the drug [103]. IOP was tested in albino rabbits; its maximum decrease was 29.5% less than the initial values, and its duration was four days. Subsequently, Kulkarni et al. [120] proposed a controlled-release ocular film of TM using a natural hydrogel derived from the seeds of *Tamarindus indica.* In three rabbits, they obtained a controlled drug release for 24 h in vivo. A self-assembling elastin-like hydrogel was experimented by Fernańdez-Colino and co-workers. In vivo testing revealed a hypotensive effect lasting more than an eighth [114]. In 2017 Karavasili et al. tested self-assembling peptides Ac-(RADA)_4_-CONH_2_ and Ac-(IEIK)_3_I-CONH_2_, which form hydrogels in physiological conditions, as carriers for ocular delivery of TM [121]. Ac-(RADA)-CONH was demonstrated to significantly enhance the bioavailability of TM, achieving effective IOP reduction for up to 24 h [121].

Dubey and colleagues prepared a stimuli-sensitive hydrogel with Carbopol (poly(acrylic acid)) [122]. TM and BT were used in this formulation, propyl methylcellulose as a viscosity enhancer, and ethylenediaminetetraacetic acid (EDTA) as a chelating agent. This delivery system achieved a 100% cumulative release of the drugs in 8 h and a higher IOP reduction efficiency in the in vivo studies [122]. In the same way, the use of carbopol was reported by Singh et al. along with hydroxyethyl cellulose for TM delivery [123]. The in vitro results showed a continuous release of the drug over eight hours and a gradual decrease of IOP if compared to marketed eye drops [123]. Subsequently, El-Feky et al. developed an oxidized sucrose crosslinker used in the formulation of chitosan-gelatin hydrogel for the sustained release of TM to control ocular hypertension [108]. They obtained a hypotensive effect that lasted more than eight hours [108].

Pakzad et al. synthesized a chitosan with Glycidyltrimethylammonium chloride (GTMAC) to prepare N-(2-hydroxy-3-trimethylammonium) propyl chitosan chloride (HTCC) [124]. HTCC increased mucoadhesive capacity, solubility in water, and antibacterial properties at physiological pH. A hydrogel was prepared with HTCC and glycerolphosphate. The studies showed an extended-release of TM for up to 80 h [124]. An in situ gel forming self-assembling peptide, ac-(RADA)_4_-CONH_2_, was evaluated as a carrier for the ocular co-delivery of TM and BT by Taka et al. [105]. A complete release of both drugs was obtained within eight hours. A 5.4-fold and 2.8-fold higher corneal permeability was achieved for BT and TM, respectively [105]. Patel et al. synthesized a combined pH and ion-sensitive in situ hydrogel using Carbopol and gellan gum, which compared with TM’s marketed eye drop formulation [104]. Furthermore, they evaluated the efficacy of benzododecenium bromide as a preservative and a corneal permeability enhancer [104]. The novel formulations reported better corneal permeability and fewer side effects than the marketed formulation [104].

In 2021, Wang et al. tested the delivery efficacy of BT and TM loaded into dendrimer gel particles of various sizes: nano-in-nano dendrimer hydrogel particles of 200 nm (nDHP) and two micronized DHPs—μDHP3 (3 μm), and μDHP10 of (9 μm) [125]. They evaluated nDHP superiority in cytocompatibility, corneal permeability, degradability, and drug release kinetics. Indeed, μDHP10 and μDHP3 induced a cytotoxicity 2.8-fold higher than nDHP, while they had a faster degradation time. Moreover, nDHP enabled 4.1 μg of BT to permeate through the cornea, while μDHP10 and μDHP3 were able to permeate, respectively, 2.4 μg and 3.5 μg. Finally, nDHP showed a longer release time [125]. The mean IOP lowering after treatment was 18.68 ± 1.35 mmHg [125]. Another research proposed a new multilayered drug delivery hydrogel inspired by a lollipop structure. The final polymer was a multilayered sodium alginate-chitosan hydrogel ball decorated by zinc oxide-modified biochar, encapsulating TM and levofloxacin inside the different layers. The results showed that in vitro release of TM can be sustained for longer than two weeks [126].

#### 2.2.3. Brimonidine Tartrate

Brimonidine tartrate exerts its effects in the eye due to its high a_2_- adrenoceptor affinity, which is considered a standard reference compound for the treatment of glaucoma [127]. As stated above, several papers reported hydrogel-based delivery strategies evaluated for the combination of BT and TM [99,105,119,122,126]. In 2017 Fedorchak et al. proposed an innovative drug delivery system composed of a thermo-responsive hydrogel carrier and BT-loaded poly(lactic-co-glycolic) acid microspheres [128]. Their results suggest in vivo efficacy for over 28 days from a single drop instillation [128]. Another paper reported a mildly cross-linked dendrimer hydrogel synthesized through the addition of a polyamidoamine (PAMAM) dendrimer and polyethylene glycol diacrylate (PEG-DA) [129], resulting in a 48 h drug release and an enhanced corneal permeation [129]. Subsequently, Wang et al. developed a branched polyrotaxane hydrogel made of 4-arm polyethylene glycol (4-PEG) and a-cyclodextrin (a-CD) [130]. BT was loaded on the resulting a-CD/4-PEG hydrogel, which underwent a reversible gel-sol transition in response to shear stress change [130]. A controlled release of 24 h was obtained. In 2019 Bellotti and co-workers reported a pNIPAAm-based thermoresponsive hydrogel for BT delivery [131]. They manipulated gelation kinetics by modifying the poly(ethylene glycol) content, thus obtaining a suitable viscosity for administration as an eye drop and resisting various climatic conditions without being eliminated [131].

#### 2.2.4. Latanoprost

Latanoprost is an ester prodrug analog of prostaglandin and is the first-line treatment in patients suffering from glaucoma because it reduces IOP by increasing uveoscleral outflow [132]. However, its daily administration is correlated with local side effects such as conjunctival hyperemia and dry eye syndrome [97]. Several polymers have been developed as drug carriers to prolong Latanoprost permanence on the ocular surface and minimize the possibility of therapy adherence failure.

In 2014, Cheng and co-workers proposed an injectable thermosensitive chitosan/gelatin/glycerol phosphate (C/G/GP) hydrogel as a sustained-release system of latanoprost for glaucoma treatment [98]. In vivo evaluation, performed on ten albino rabbits, showed a decrease of IOP of 9.2% within eight days and then a permanence within normal limits for the next 31 days [98]. Hsiao et al. prepared a chitosan-based thermogel to improve glaucoma therapy [106]. In vivo examination, conducted with subconjunctival injections in six rabbits, revealed excellent biocompatibility and a lowering of IOP for 40 days [106].

In the same way, Cheng et al. formulated a thermosensitive chitosan/gelatin for the sustained release of latanoprost as a topical eye drop to control ocular hypertension [133]. In vivo results of a rabbit model confirmed IOP was significantly decreased within seven days [133]. In 2019, another thermosensitive hydrogel containing latanoprost and curcumin-loaded nanoparticles was developed [97]. Indeed, curcumin possesses antioxidant and anti-inflammation properties and reduces oxidative stress on trabecular meshwork [97]. In vitro drug release evaluation revealed that both latanoprost and curcumin-loaded nanoparticles maintained a sustained release for seven days [97].

#### 2.2.5. Other Drugs

Epinephrine was applied in glaucoma treatment to reduce intraocular pressure by decreasing aqueous formation and increasing the outflow facility [134]. Hsiue et al. developed two preparations of ophthalmic drops for controlled release based on the thermosensitivity of poly-N-isopropyl acrylamide (PNIPAAm) [135]. The first formulation contained a linear chain of PNIPAAm, and the second had a mixture of linear PNIPAAm and cross-linked PNIPAAm nanoparticles. After in vivo examination of the rabbits, the authors reported that the formulation containing a linear PNIPAAm and nanoparticles maintained a more prolonged IOP decrease (32 h) but showed a weaker effect. In contrast, the linear PNIPAAm, had a shorter duration but a more substantial IOP decrease (−7.2 mmHg) [135]. In the same way, Prasannan and colleagues structured a thermosensitive PNIPAAm-based hydrogel loaded with epinephrine [136].

Atenolol is a β1 adrenoceptor blocker for the treatment of glaucoma. A niosomal hydrogel containing atenolol was proposed by Abuhashim et al. in 2014 [137]. Their formulation was an eye drop solution that showed promising results after in vivo examination as it significantly decreased the IOP and showed a prolonged effect up to 8 h [137].

**Table 2 gels-08-00510-t002:** Studies on drug delivery hydrogel.

Author	Year	Drug	Group	Polymer Name	IOP Decrease (Mean Value)	Duration	Administration
Bellotti et al. [131]	2019	Brimonidine tartrate	α2 agonist	pNIPAAm hydrogels (PEG)	/		Eye drop
Fedorchak et al. [128]	2017	Brimonidine tartrate	α2 agonist	Poly(lactic-co-glycolic) acid microspheres microspheres incorporated into the pNIPAAM gel	/	28 days	Eye drop
Wang et al. [129]	2017	Brimonidine tartrate	α2 agonist	Linked dendrimer hydrogel via addition of polyamidoamine (PAMAM) dendrimer G5 and polyethylene glycol diacrylate (PEG)	/	48 h	AC filling
Wang et al. [130]	2018	Brimonidine tartrate	α2 agonist	(a-CD/4-PEG hydrogels) hydrogel made of 4-arm polyethylene glycol (4-PEG) and a-cyclodextrin (a-CD)	/	24 h	/
Dubey et al. [122]	2014	Timolol maleate- brimonidine tartrate	β-blockers- α2 agonist	Stimuli-sensitive hydrogel with Carbopol (poly(acrylic acid)	14 mmHg (mean IOP after treatment)	8 h	Eye drop
Holden et al. [119]	2012	Brimonidine-timolol maleate	α2 agonist- β-blockers	Polyamidoamine dendrimer hydrogel linked with polyethylene glycol (PEG)-acrylate chains	/	6–72 h	/
Taka et al. [105]	2020	Timolol maleate-brimonidine tartrate	β-blockers- α2 agonist	Self-assembling peptide ac-(RADA)4-CONH2	/	8 h	/
Wang et al. [125]	2021	Brimonidine tartrate-and timolol maleate	α2 agonist β-blockers	Nano-in-nano dendrimer hydrogel particles −200 nm (nDHP)	18.68 ± 1.35 mmHg (mean IOP after treatment)	/	Eye suspension
Yang et al. [99]	2013	Brimonidine-timolol maleate	α2 agonist- β-blockers	Hybrid dendrimer hydrogel/poly(lactic-co-glycolic acid) nanoparticle platform	29.5%	4 days	/
Cheng et al. [133]	2016	Latanoprost	Prostaglandin	Thermosensitive chitosan/gelatin	/	7 days	Eye drop
Cheng et al. [97]	2019	Latanoprost	Prostaglandin	Thermosensitive hydrogel containing latanoprost and curcumin-loaded nanoparticles	/	7 days	Eye drop
Cheng et al. [98]	2014	Latanoprost	Prostaglandin	Thermosensitive chitosan/gelatin/glycerol phosphate (C/G/GP) hydrogel	2.4 mmHg (9.2%)	31 days	Subconjunctival injection
Hsiao et al. [106]	2014	Latanoprost	Prostaglandin	Amphiphilic chitosan-based thermogelling	10 mmHg	39 days	Subconjunctival injection
Abu Hashim et al. [137]	2014	0.5% atenolol	β1 adrenoceptor blocker	Niosomal Hydrogel containing atenolol	/	8 h	Eye drop
Hsiue et al. [135]	2002	epinephrine	Catecholamine	Thermosensitive poly-N-isopropylacrylamide (PNIPAAm)	8.9 mmHg (maximum)	24 h	Eye drop
Prasannan et al. [136]	2014	epinephrine	Catecholamine	PAAc-g-PNIPAAm (PNIPAAm)	/	/	Eye drop
Chou et al. [107]	2017	pilocarpine	Cholinergic	Pilocarpine-loaded gallic acid (GA)-grafted gelatin-g-poly(N-isopropylacrylamide) (GN)	5 mm Hg	28 days	/
Lai et al. [111]	2013	pilocarpine	Cholinergic	Carboxyl- terminated PNIPAAm	/	12 h	/
Luo et al. [113]	2020	pilocarpine and RGFP966	Cholinergic	4-hydroxy-3,5-dimethoxybenzoic acid (p-DMB)-modified chitosan-g-poly(N-isopropylacrylamide)	/	70 days	/
Natu et al. [110]	2007	Pilocarpine hydrochloride	Cholinergic	Linking reaction of gelatin in N,N-(3dimethylaminopropyl)-N′-ethyl carbodiimide and N-hydroxy succinimide	/	8 h	/
Nguyen et al. [112]	2019	Pilocarpine and ascorbic acid	Cholinergic	PAMAM dendrimers bearing amine surface groups (-NH2) linked with gelatin hydrogel and poly(N-isopropyl acrylamide)	/	84 days	/
El-Feky et al. [108]	2018	Timolol Maleate	β-blockers	Chitosan-gelatin hydrogel linked with oxidized sucrose	/	8 h	Eye drop
Esteban-Pérez et al. [117]	2020	Timolol maleate	β-blockers	Gelatin nanoparticles in a hydroxypropyl methylcellulose viscous solution	4.33 ± 0.30	8 h	Eye drop
Fernandez-Colino et al. [114]	2017	Timolol maleate	β-blockers	Self-assembling elastin-like (EL) and silk-elastin-like hydrogels	/	8 h	Eye drop
Karavasili et al. [121]	2017	Timolol maleate	β-blockers	Self-assembling peptides Ac-(RADA)4-CONH2 and Ac-(IEIK)3I-CONH2	/	24 h	Eye drop
Kulkarni et al. [120]	2016	Timolol maleate	β-blockers	Natural hydrogel from Tamarindus indica	/	24 h	Eye drop
Pakzad et al. [124]	2020	Timolol maleate	β-blockers	N-(2-hydroxy-3-trimethylammonium) propyl chitosan chloride glycerophosphate (HTCC/GP)	/	1 week	/
Wang et al. [126]	2021	Timolol maleate (TM) and levofloxacin	β-blockers	Multilayered sodium alginate-chitosan (SA-CS) hydrogel ball (HB) decorated by zinc oxide-modified biochar (ZnO-BC) (‘lollipop inspired’)	/	2 weeks	/
Zhang et al. [118]	2011	Timolol maleate	β-blockers	Liposomal-hydrogel	/	6 h	Eye drop

## 3. Hydrogel Formulation after Glaucoma Surgery

### 3.1. Anti-Scarring Hydrogel

Glaucoma filtration surgery is currently the most effective treatment for glaucoma unresponsive to medical therapy. Filtering surgery lowers IOP by forming a fistula between the anterior chamber and the subconjunctival space, favoring the drainage of aqueous humor in a filtering bleb. However, traditional trabeculectomy and filtering MIGS fail in a significant percentage of patients, up to 30–50%, mainly because of fibrosis in the subconjunctival filtering bleb [138,139].

The reparative process, especially in the first two weeks after surgery, involves the release of a great variety of proinflammatory cytokines and glycoproteins of the extracellular matrix. The latter stimulates the migration and proliferation of fibroblasts in the Tenon’s capsule and the subconjunctival space leading to the formation of a scar [140].

For this reason, glaucoma filtration surgery involves using antifibrotic drugs applied intraoperatively for a variable time (generally between 2 and 5 min). The most used drugs are mitomycin C (MMC) and 5-fluorouracil (5-FU), but cyclosporine A or antibodies against vascular endothelial growth factor (VEGF) such as Bevacizumab are also used. However, MMC and 5-FU, in particular, have a non-specific mechanism of action for which, in addition to inhibiting fibroblastic proliferation, they also induce cell death in the surrounding tissues giving rise to potential complications (postoperative hypotonia, bleb leaks, epithelial and endothelial corneal toxicity, thinning up to the rupture of the bleb with the consequent risk of endophthalmitis) [141,142,143,144]. Furthermore, a single application of these drugs may not be sufficient to limit the fibrotic and connective tissue reaction that develops even in the long term after the surgery [145].

Therefore, in recent years delivery systems have been studied to allow a controlled and prolonged release of drugs over time to limit the formation of postoperative scar, keep the IOP low, and guarantee the persistence of the filtering bleb while reducing toxicity. As stated above, nanoparticles-based hydrogels and stimuli-responsive hydrogels, for their natural properties, are the best candidates to play the role of drug carrier [146,147]. Their simplicity of synthesis has the advantage of precisely controlling the concentration of the incorporated drug, which is mixed in the initial aqueous solution, and of not denaturing peptides or bioactive proteins within them [148,149]. The porous structure of hydrogels allows them to transport many therapeutic molecules, both hydrophilic and hydrophobic, such as proteins, DNA, and RNA.

The hydrogels are degraded slowly, in a variable way, between two and three weeks. This characteristic makes them ideal for transporting and releasing anti-scarring drugs that can act with a stable and effective concentration precisely in the critical period of scar tissues [150].

Table 3 lists the primary studies on using anti-scarring hydrogels in glaucoma filtration surgery.

Some studies focus on the intrinsic hydrogel antifibrotic activity. In the works of Liang et al. [151] and Martin et al. [152], these biomaterials, if applied to the intervention site, form covalent bonds with the sclera and significantly reduce scar formation. Hydrogels minimize inflammation and hinder fibroblasts’ adhesion to scleral tissue, as confirmed by the reduction of connective tissue growth factor (CTGF), a peptide promoting the formation of fibrosis and scarring in ocular tissues [153]. Chen et al. [154] investigated in vitro hydrogels containing different concentrations of the arginine-glycine-aspartic acid (RGD) sequence (0.25; 0.5 and 1%). This sequence induces competition with proinflammatory proteins of the extracellular membrane for binding to integrins, cellular receptors that promote migration, and the proliferation of fibroblasts without cytotoxicity. The authors could demonstrate that 1% wt Fmoc-FFGGRGD self-assembly peptide hydrogel could inhibit the expression of β1-integrin, FAK, and Akt in Tenon’s capsule fibroblasts, which play an essential role in fibrogenesis and scar formation.

The majority of authors used hydrogels as a drug delivery system: they are implanted and provide for a localized and prolonged release of the drug they contain. Nagata et al. [155], Xi et al. [156], and Kojima et al. [157] studied MMC-loaded hydrogels. They demonstrated that hydrogel causes a prolonged and controlled release of MMC, reducing its toxicity while simultaneously reducing the formation of fibrosis and ensuring a prolonged persistence of the bleb. Yang et al. [158], Peng et al. [159], and Han et al. [150] used hydrogels to release Bevacizumab, a synthetic monoclonal antibody directed against VEGF. This drug, widely used in proliferative diabetic retinopathy, age-related macular degeneration, and neovascular glaucoma, limits the formation of fibrosis and scar after glaucoma filtration surgery and reduces IOP. VEGF is a crucial molecule in the wound healing process: it is not only an essential promoter of angiogenesis but also a direct mediator of the migration and proliferation of fibroblasts and inflammatory cells [160].

Other authors investigated the release of less commonly used drugs in glaucoma filtration surgery to reduce toxicity. Kojima et al. [161] studied the use of a chymase inhibitor. Chymase is a protease in the granules of mast cells that induces the accumulation of neutrophils, eosinophils, and other inflammatory cells and promotes cell growth of fibroblasts through the up-regulation of transforming growth factor (TGF-β) [162,163]. Using the chymase inhibitors could prevent scar formation and cause fewer complications than the current antimetabolites. Maeda et al. [164] directly associated a TGF-β antibody with the hydrogel. TGF-β is the main factor stimulating the conjunctival scar following trabeculectomy [165]. Sun et al. [166] used Cyclosporine A, while Qiao et al. [167,168] used heparin, an anticoagulant that also can limit the proliferation of fibroblasts. Chun et al. [169] used the hydrogel as a vector of small interfering RNA (siRNA). This promising therapy acts through an epigenetic silencing strategy, deactivating the genes that code for proteins promoting fibrosis and scar formation.

Another area of investigation is the administration site of the hydrogel. Most studies involve a subconjunctival injection in correspondence with the filtering bleb, and other authors suggest applying the hydrogel under the scleral flap created during the surgery [170]. A further strategy is the anterior chamber implant, as proposed by Peng et al. [159] and Han et al. [150]. The intracameral injection allows a controlled and stable release of the drug in the anterior chamber and, at the same time, in the filtering bleb through the drainage of aqueous humor determined by the surgery, without further manipulation of the conjunctiva [171].

All the studies reported favorable data regarding the reduction of fibrosis and scar formation and low toxicity. Hydrogels could therefore represent an excellent potential to increase the long-term success of these interventions.

### 3.2. Management of Others Post or Intraoperative Complications

One of the most frequent complications in the early postoperative period after glaucoma surgery is the leakage from the conjunctival limbal incision or the filtration blebs [172]. This leakage can lead to severe complications such as hypotony, choroidal effusion, suprachoroidal hemorrhages, loss of the anterior chamber, endophthalmitis, and bleb failure [173,174]. The leakage occurrence as a short-term complication after glaucoma surgery is becoming more frequent because of the increased application of mitomycin-C to delay wound healing [175]. Nagata et al. proposed a PEG-Based Synthetic Hydrogel as a sealant after glaucoma surgery to inhibit bleb leaking [155]. On a rabbit model, they observed fewer lymphocytic infiltrations without inflammatory effects or conjunctival toxicity [155].

Calladine et al. proposed an intraoperative implant of methacrylic hydrogel, applicable during deep sclerectomy, to maintain the intrascleral space essential for proper filtration after this surgery [176]. Their polymer showed good intrascleral biocompatibility, while no case of hypotony was reported [176].

## 4. Conclusions

The global cost of sight loss is estimated to be over US$3 trillion annually. Glaucoma is the second-leading cause of irreversible visual loss and is mainly treated with eye drops. Although the drug delivery formulations represent a promising alternative to conventional treatment, there are still limitations related to the performances of these ocular delivery systems. Poor adherence and/or persistence with topically applied eye drops mainly results from the need for multiple daily applications to obtain its intended therapeutic effect. New 3D-printed hydrogels, ophthalmic gels, medical devices, and nanogels designed to deliver ophthalmic gels have the potential to mitigate glaucoma-related comorbidities. Although numerous studies have demonstrated the great potential of hydrogel-based treatments, future research should continue to investigate their use in vivo and conduct clinical trials to advance the clinical application of this technology.

Furthermore, modifications of traditional surgery and the introduction of new devices to shunt aqueous humor subconjunctivally significantly reduced the early postoperative complications related to hypotony but are still facing a significant failure rate due to fibrosis of the bleb. The development of longer-releasing antifibrotic agents that tackle the different phases of the scarring process would allow for a longer-term efficacy of surgery and may potentially allow an earlier surgical approach with better control of IOP throughout the day.

## Figures and Tables

**Figure 1 gels-08-00510-f001:**
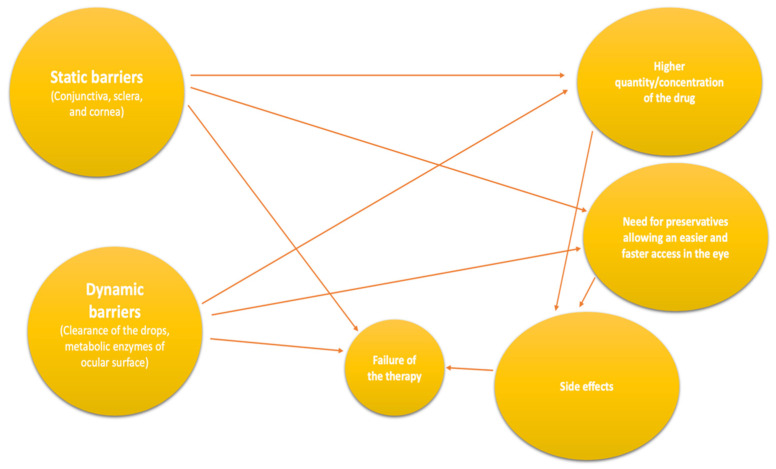
Causes of therapy failure.

**Figure 2 gels-08-00510-f002:**
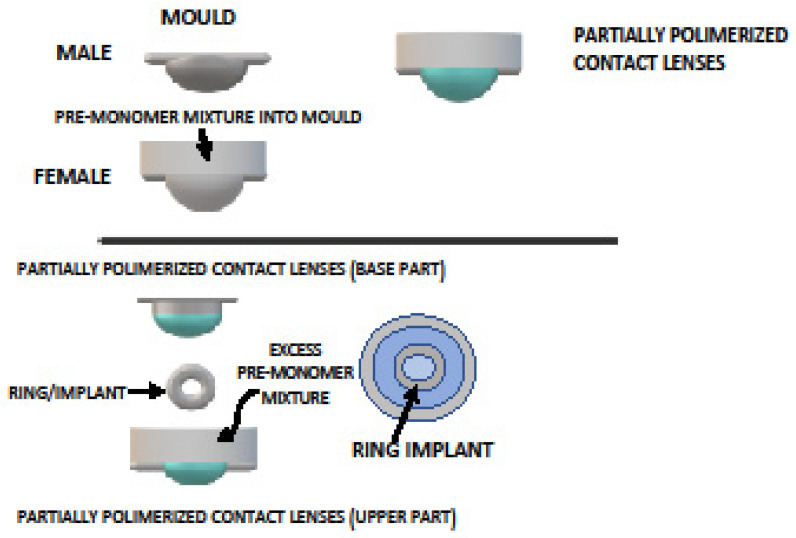
A nanoparticle-loaded ring implant placed between partially polymerized hydrogel CL.

**Table 1 gels-08-00510-t001:** Studies on contact lens hydrogel.

Author	Year	Drug	Group	Polymer Name	Manufacturing	IOP Decrease	Duration
Maulvi F.A. [82]	2021	Bimatoprost	Prostaglandin	Graphene oxide	Bimatoprost before polymerization	/	
Garcia Fernandez M.J. [93]	2013	Ethoxzolamide	CAI	Poly-HEMA and poly-HEMA-co-APMA	Dry disk immersed into	/	10 days
Jung H. J. [80]	2012	Timolol	B-blockers	Propoxylated glyceryl triacylate	Adding particles to the polymerization mixture	/	<=5 days
Maulvi F.A. [81]	2016	Timolol	B-blockers	Ethyl cellulose nanoparticle-laden ring	TM loaded ring in hydrogel contact lens	Decrease by 6.3 ± 1.92 mmHg after three hours	8 days (in vivo)
Mohammadi S. [94]	2014	Latanoprost	Prostaglandin	Balaficon A/Senofilcon A	Incubation in drug solution		>24 h
Peng C. C. [95]	2012	Timolol	B-blockers	NIGHT&DAY silicone hydrogel contact lenses With/without vit. E	Soaked	Decreased by 5	/
Ciolino J.B. [55]	2016	Latanoprost	Prostaglandin	Methafilicon+ methacrylic acid (Hydrogel)	Photopolymerization	Low dose: decreased > 6. High dose decreased> 10	/
Sekar P. [84]	2019	Bimatoprost and Latanoprost	Prostaglandin	Vit E added to ACUVUE OASIS and ACUVUE TRUE EYE	Soaked	/	>10 days
Yan F. [59]	2020	Bimatoprost	Prostaglandin	HEMA (hydroxyl ethylmethacrylate)	Imprinting vs. soaked	/	Imprinted 36–60 h
Xu J. [64]	2010	Puerarin	Chinese medicine ability to block b-receptors	pHEMA-NVP-MA	Soaked	/	350 min

**Table 3 gels-08-00510-t003:** Studies on anti-scarring hydrogels in glaucoma filtration surgery.

Author	In vitro/Vivo	Hydrogel	Function	Drug Delivered	Activation Mode	Administration Site
Blake et al. J Glaucoma, 2006 [145]	in vitro	P(HEMA)	Drug delivery system	Mitomycin C	/	/
Liang et al. Biomed Mater., 2010 [151]	in vivo	Peptide hydrogel with RGD sequence	Keeping tissues apart, inflammatory inhibition	/	/	Filtering bleb
Yang et al. Acta Pharmacol Sin., 2010 [158]	in vitro and vivo	CMCS	Drug delivery system	5-fluorouracil, bevacizumab	/	Filtering bleb
Kojima et al. Invest Ophthalmol Vis Sci., 2011 [161]	in vivo	Gelatin-hydrogel	Drug delivery system	Chymase inhibitor	/	Filtering bleb
Xi et al. PLoS One., 2014 [156]	in vitro and vivo	PTMC_15_-F127-PTMC_15_	Drug delivery system	Mitomycin C	Body temperature	Filtering bleb
Peng et al. Med Hypothesis Discov Innov Ophthalmol., 2014 [159]	in vivo	PECE	Drug delivery system	Bevacizumab	Body temperature	Anterior chamber
Han et al. J Mater Sci Mater Med., 2015 [150]	in vitro and vivo	PECE	Drug delivery system	Bevacizumab	Body temperature	Anterior chamber
Kojima et al. Invest Ophthalmol Vis Sci., 2015 [157]	in vivo	Gelatin-hydrogel	Drug delivery system	Mitomycin C	/	Filtering bleb
Sun et al. J Mater Chem B., 2017 [166]	in vitro and vivo	PLGA-PEG-PLGA	Drug delivery system	Cyclosporine A	Body temperature	Filtering bleb
Qiao et al. J Mater Sci Mater Med., 2017 [167]	in vivo	HECTS	Drug delivery system	Heparin	UV irradiation	Under scleral flap
Maeda et al. Int J Mol Sci., 2017 [164]	in vivo	Gelatin-hydrogel	Drug delivery system	TGF-β antibody	/	Filtering bleb
Martin et al. Macromol Rapid Commun., 2020 [152]	in vitro	DMAA + AOAQ	Fibroblast cells repellent	/	UV irradiation	/
Chen et al. J Biomed Mater Res B Appl Biomater., 2021 [154]	in vitro	Peptide hydrogel with RGD sequence	Peptide competition on protein binding site	/	/	/
Chun et al. Sci Rep., 2021 [169]	in vitro and vivo	Gelatin-tyramine	Drug delivery system	siRNA	Charge tunability	Filtering bleb

## Data Availability

Not applicable.

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
