# Peer review of "Glaucoma Treatment and Hydrogel: Current Insights and State of the Art"

_gels, 2022, doi:10.3390/gels8080510_

Round 1
Reviewer 1 Report
Article is good , well prepared but need few corrections:
1. Check grammar
2. Font type should be uniform including table.
3. Invitro and invivo words should be in italics throughout.
4. There is no figure, some content could be converted to a figure.
Author Response
Thank you for your kind review, we followed your suggestions and we hope this may make our paper more attractive.
Reviewer 2 Report
Detailed Review Comments
Recent uses of hydrogels in the treatment of glaucoma are compiled in the present review publication written by Fea et al. The authors have discussed the distribution methods for a number of eye medications that are frequently used for the treatment of glaucoma. The tables display all year-by-year updates and their results. The review is succinct and includes all the details for very particular reading groups. I wish to share some observations with you below.
1. "Application of hydrogel in glaucoma" is in the review's title. Section 2.2, however, is structured differently. It would be excellent if hydrogel materials were highlighted because many people would find this review to read about the many hydrogels used so far for ocular medication delivery. Therefore, I propose that mention the type of hydrogel/polymer be employed and, in that regard, which drugs are being released might be specified, rather than having drugs eg. Pilocarpine as "2.2.1" and so on. Alternately, the title might change.
2. In the continuity of my first comment I did a google search for the keyword ‘glaucoma and hydrogels” I am sharing the links of the reviews which came as a result.
a. Frontiers | Considerations for Polymers Used in Ocular Drug Delivery (frontiersin.org)
b. Intravitreal Injectable Hydrogels for Sustained Drug Delivery in Glaucoma Treatment and Therapy - PMC (nih.gov)
c. Hydrogel Biomaterials for Application in Ocular Drug Delivery - PMC (nih.gov)
d. Recent advances in drug delivery systems for glaucoma treatment - ScienceDirect
3. I found the current review different in terms of organization. The current review is organized by drugs delivered for glaucoma treatment. So title change could be considered.
4. Despite having excellent writing, the review lacks visual images. It is requested that the author condense all the important details (including the various hydrogel types, medication types, and glaucoma/eye representative photos) into a single image and include at least one figure.
I am confident that incorporating the aforementioned adjustments will raise the manuscript's caliber and be advantageous to the reader. In conclusion, the article is suited for publication in Gels.
Author Response
1., 2., 3.
Thank you for your suggestion. Our paper was meant to be mainly for clinicians, thus our decision to structure the essay this way. Changing the whole structure would be too difficult, and as you pointed out, other relatively recent papers tackle the problem from a hydrogel-polymer point of view. Thus, we opted to change the title to "Glaucoma Treatment and Hydrogel: "Current Insights and State of the Art," hoping that this will meet the interest of clinicians and material scientists.
Furthermore we added a chapter called ‘Polymers and Hydrogel in Ocular Drugs Delivery’ adding references you suggested with the following text:
“Hydrogels are a network of biologic and synthetic hydrophilic polymers able to absorb aqueous fluids. The large array of customizability and the high hydrophilicity, render hydrogel the ideal compound for ophthalmic applications, ranging from vitreous substitutes to drug delivery [21]. Several authors proposed hydrogel as drug carrier, incorporated with other polymers, for the treatment of glaucoma. Among these molecules, synthetic polymeric biomaterials are a group of compounds based on chemically derived monomers and represent one of the most reported options for the delivery of ocular pharmaceuticals [21]. To this class belong polyethylene oxides, polyvinyl alcohol (PVA), polyesters, polymethacrylates, polyolefins and dendrimers [22]. On the contrary, biopolymers are based on naturally derived monomers and are characterized by a high biocompatibility and a fast degradation [23]. The most frequently reported biological polymers in use for ocular drug delivery are chitosan, cellulose, hyaluronic acid, carboxymethyl cellulose and gelatin. Another promising field as ocular delivery systems are micro and nanotechnology such as formulations of hydrogel combined with micro- and nanoparticles, liposomes and micelles [22]. This drug delivery strategy combines the hydrogel’s high-water content to the small size (from 1 to 1000 nm) of these particles, offering a more targeted delivery and a sustained release of the medications [23].
Among the medical gels, the most investigated are the gel-forming systems. Their properties make them able to undergo phase transition after environmental stimulus following their application in the conjunctival cul-de-sac and thus, transform themselves into a viscoelastic gel from the original liquid dosage [24] because of pH change, temperature modulation, or ion triggers [25][26][27][28][29][30]. Because they don't need organic solvents or copolymerization catalysts to cause gel formation, in situ polymeric gelling systems, have drawn significant attention.”
This way we explained the categories of hydrogels used for drugs delivery.
4.
Thank you, adding pictures from other publications would have certainly been important. Nevertheless, this will imply the permission of the Journal and will take some time. We decided, thus, to add some explanatory illustrations.
Reviewer 3 Report
Fea et al. report an overview of the current applications of hydrogels in glaucoma. They review current applications of ophthalmic gelling systems with particular emphasis on glaucoma. However, several issues have been found before further consideration of publication:
1. What is HEMA hydrogel?
2. Hard or soft contact lense should be discussed
3. It should be realized that evem water content of the lens by e.g., poly HEMA, etc., reaches 99%, the oxygen permeability (DK/t)will not exceed the theoretical value of about 40 that affect the user comfort during prolonged period of time.
4. Contact lense made of PDMS hydrogel and their drug release should be discussed
5. Representative figures from other work should be included for the glaucoma treatment
6. The author should discuss whether the physical/mechanical properties of the lense affect the drug delivery efficiency
7. It is quite disappointing that dexamethasone/glucocorticoids has not been discussed for treating glaucoma
8. Table word sizes and styles are different from the text
9. A lot of extra spaces and grammatical mistakes are detected.
Author Response
1.
Thank you, we explained the acronym. P HEMA is Poly (2-hydroxyethyl methacrylate).
2.
Thank you for your observation. Nevertheless, the paper is meant to be on gels; we are unaware of any hard lens using gel material. We will be happy to add it if you would be so kind as to provide any references.
3.
Thank you, we have added the sentence:
"Indeed, even if the water content of the lens reaches 99%, the oxygen permeability will not exceed the theoretical value of 40 DK/t affecting the user comfort during a prolonged time."
4.
Thank you. This is an observation that can add some interest to the paper. Thus, we added the text:
"Nicolson et Al. found out that hydrogel lenses based on hydrophilic monomers such as 2-hydroxyethyl methacrylate (HEMA) and N-vinyl pyrrolidone (NVP) did not succeed in absorbing the minimum oxygen requirement for ocular eyes under the closed eyelid conditions."
"Nguyen-Phuong-Dung et al. show that higher content of hydrophilic polymers increased water uptake ability and improved hydrophilicity of silicone hydrogel lenses. However, the oxygen permeability is linked with the quantity of PDMS: the permeability decreases with the decrease of PDMS content. In addition, these silicone hydrogel lenses exhibited relatively optical transparency, anti-protein deposition, and appeared to be non-cytotoxic."
"Chen et al. and Jones et Al. explain that PDMS exhibits deficiencies of low water uptake ability, poor wettability, and high lipid adsorption despite having unique high oxygen transmissibility and superior resistance to tearing for contact lenses application."
5.
Thank you, adding pictures from other publications would have certainly been important. Nevertheless, this will imply the permission of the Journal and will take some time. We decided, thus, to add some explanatory illustrations.
6.
Thank you. We introduced the chapter on contact lenses explaining the physical advantages of CL: “Due to their location near the cornea, contact lenses (CL) have some unique advantages for delivering drugs [42]. First of all, the limitation opposed by the CL to the tear film action results in a drug residence time of more than 30 min [43][44] compared to 5 min only for eye drops alone [45]. The enhanced residence time leads to significant increases in bioavailability, possibly as large as 50% [46]. Notably, hydrogels are the main constituents of CL; thus, their high-water content and favourable properties render them highly compatible with human tissues [47][48][49].”
Then following your suggestion we added this sentence that provide an important information: "Indeed, even if the water content of the lens reaches 99%, the oxygen permeability will not exceed the theoretical value of 40 DK/t affecting the user comfort during a prolonged time."
Then we concluded the text with this information providing a further explanation about physical and chemical properties and limits of the lens: “One advantage of the hydrophobic drug released from CL is that it could more readily diffuse through corneal cells' tight lipid junctions and cell membranes due to its hydrophobic character [93]. Furthermore, different studies have found that drugs with hydrophobic character (ex. Latanoprost) can be loaded quickly (in 4 minutes) into Hydrogel contact lenses using non-aqueous solvents [93][94]. One of the most critical problems of drug delivery through CL remains the difficulty of incorporating more than one drug. A high degree of cross-link hydrogel is needed in this case, but this process might alter physical properties or oxygen permeability [95] (Table 1).”
7.
Thank you for this observation: hydrogels are essential for carrying steroids. Nevertheless, steroids are used marginally in glaucoma therapy, mainly for uveitic glaucoma. Even in this case, though, only a few conditions may benefit from using hydrogels carrying steroids. In most cases, the therapy is mainly based on eye drops. Contrary to primary open or closed angle glaucoma, the treatment of uveitic glaucoma is limited to the burst of uveitis. Thus we decided not to include a part on steroids in this review.
8.
Thank you, we have corrected and we uniformed all the styles in the article.
9.
Thank you, we carefully went through the text, and we hope to have resolved these criticisms.
Round 2
Reviewer 3 Report
The authors have addressed the comment appropriately